# Design and Test of a High-Performance Wireless Sensor Network for Irradiance Monitoring

**DOI:** 10.3390/s22082928

**Published:** 2022-04-11

**Authors:** Manuel Jesús Espinosa-Gavira, Agustín Agüera-Pérez, José María Sierra-Fernández, Juan José González de-la-Rosa, José Carlos Palomares-Salas, Olivia Florencias-Oliveros

**Affiliations:** Research Group PAIDI-TIC-168, Computational Instrumentation and Industrial Electronics (ICEI), University of Cadiz, ETSI of Algeciras, Av. Ramonn Puyol S/N, Algeciras, 11202 Cadiz, Spain; agustin.aguera@uca.es (A.A.-P.); josemaria.sierra@uca.es (J.M.S.-F.); juanjose.delarosa@uca.es (J.J.G.d.-l.-R.); josecarlos.palomares@uca.es (J.C.P.-S.); olivia.florencias@uca.es (O.F.-O.)

**Keywords:** irradiance WSN, high-performance WSN, solar irradiance, low-cost WSN, online wireless sensor network, irradiance network, nowcasting

## Abstract

Cloud-induced photovoltaic variability can affect grid stability and power quality, especially in electricity systems with high penetration levels. The availability of irradiance field forecasts in the scale of seconds and meters is fundamental for an adequate control of photovoltaic systems in order to minimize their impact on distribution networks. Irradiance sensor networks have proved to be efficient tools for supporting these forecasts, but the costs of monitoring systems with the required specifications are economically justified only for large plants and research purposes. This study deals with the design and test of a wireless irradiance sensor network as an adaptable operational solution for photovoltaic systems capable of meeting the measurement specifications necessary for capturing the clouds passage. The network was based on WiFi, comprised 16 pyranometers, and proved to be stable at sampling periods up to 25 ms, providing detailed spatial representations of the irradiance field and its evolution. As a result, the developed network was capable of achieving comparable specifications to research wired irradiance monitoring network with the advantages in costs and flexibility of the wireless technology, thus constituting a valuable tool for supporting nowcasting systems for photovoltaic management and control.

## 1. Introduction

The increase in the installed photovoltaic (PV) power observed in recent years is expected to continue for both distributed and utility-scale systems [1]. With increasing penetration levels, grid stability and power quality can be affected by cloud-induced PV variability posing a series of technical challenges that have been the main focus of research in the topic during the last decade [2]. Techniques such as power curtailment or Battery Storage Energy Systems (BESS) have been widely used to minimize the high frequency PV power variability derived from cloud passage [3,4]. It has been demonstrated that forecasts in the range of seconds are necessary to take full advantage of these techniques. Concretely, power curtailment has proven to be efficient in smoothing upwards power ramps, but its application to the downward ones requires forecasts to perform curtailment before cloud arrival [5]. BESS based techniques can smooth both upwards and downwards ramps, but forecasts are also required for the optimized control of the charge/discharge cycles, which allows a reduction of battery capacity and an extension of its lifespan [6].

The term ’nowcasting’ is frequently associated with ultra-short-term forecasts to clearly differentiate it from short-term forecasting, which involves longer prediction horizons and different applications. Namely, short-term forecasting is associated with horizons from tens of minutes to some hours, with applications in load dispatch, intra-day market, and operational security of the electricity system [7]. Short-term forecasting is generally based on Numerical Weather Prediction (NWP) models and satellite observations with the goal of estimating the average cloud coverage over the period rather than the detection of individual clouds. On the other hand, nowcasting focuses on the range of seconds to few minutes, being used in real-time operation and control of PV systems and distribution feeders with significant PV penetration levels [8,9]. In this case, it is necessary to consider irradiance transitions generated by the passage of individual clouds, thereby requiring data sources capable of providing this information: satellite imagery, sky cameras, or irradiance sensor networks. However, satellite imagery has a limited spatial resolution and low sampling rate on the order of hundred of meters and several minutes, respectively. Thus, they are appropriate for the detection of atmospheric fronts that could cover the PV system in the next few minutes, but they are not particularly useful in the prediction of ramp events. Sky cameras, despite having a high acquisition rate, present difficulties in the translation of images to irradiance values [10]. Many studies focused on the reduction of errors derived from this translation, but they are generally higher than 20% Root Mean Square Error (RMSE) under partial cloud cover conditions [11,12,13].

Irradiance sensor networks are positioning themselves as a promising data source for nowcasting and short-term forecasting applications since they represent the simplest way to obtain spatial distributions of irradiance with a high sampling rate [10]. In this sense, Hoff and Perez stated that “regularly gridded networks of pyranometers” are the ideal data source for analyzing PV variability [14]. Chen et al. demanded a “dense sensor network dispersed over a wide area” as the best option to precisely capture cloud shadow features [15].

There are many examples of irradiance sensor networks in scientific literature, although they are typically designed for research purposes, with measurements stored in data loggers for an offline analysis. Table 1 includes some representative cases of these offline networks. For instance, the Oahu [16] and the Meltzip [17] networks have been used in many studies for different purposes, thanks to the data being available on their web sites. They share characteristics that are extensible to most of the offline networks: 1-s resolution, use of data-loggers, comparable areas and density of measurement points, and research focus. In Table 1, we referenced only two recent articles based on these networks for illustration purposes: data from the Oahu dataset applied to control of utility-scale PV plants [18], and data from the Melpitz dataset applied to model cloud advection [19]. The networks described in [20,21] has similar characteristics to the previous ones, although, in these cases, the pyranometers are arranged in a gridded layout.

The Singapore network used in [25] is also used as an offline data source, but, in this case, it has a more regional focus with spatial and temporal resolutions not appropriate for cloud passage events.

The above monitoring systems effectively fulfill the purpose for which they were conceived. However, in order to support real-life solutions for PV systems, irradiance sensor networks must be designed with different specifications. First, they should be able to capture and transmit data in real time or near real time. Second, the cost of the infrastructure must be economically justified. This limits the cost of the monitoring network in comparison with the PV system, as stated in [32]. In this sense, while wired networks are better in terms of latency, reliability, and data rates, they have penalties in deployment complexity, higher costs, and lower flexibility for adding new devices or moving existing ones. For instance, the wired irradiance network of the Tampere University of Technology described in [22] was able to capture 25 samples/s (sampling period of 40 ms) with 21 pyranometers connected to a CompactRIO data-acquisition system. The authors finally concluded that a sampling period of 100 ms “is sufficient for detecting even the fastest phenomena related to power generation with PV power plants”, which is the sampling frequency applied in subsequent studies based on this network [30,33]. Again, this deployment with advanced devices, complex infrastructure, and excess performance is justified by research purposes. Therefore, a wireless configuration implementing ordinary devices while meeting the necessary measurement specifications seems a more practical and cost-effective solution for real applications.

Examples of online irradiance Wireless Sensor Networks (WSN) with these characteristics are very scarce in scientific literature. There is specific research on the design of nodes for irradiance WSN [34,35], although no network is deployed in these investigations. In other cases, the potential of online irradiance WSN at regional scales is evaluated, but considering coarse spatial and temporal resolutions not adapted to cloud passage events [27,31]. In [29], the development of a rooftop irradiance WSN comprising 10 pyranometers is described, but again the 5-minutes sampling period is not adequate to deal with PV ramps.

As summary, there are only a few articles that describe real implementations of online irradiance WSN able to support control strategies for minimizing cloud-induced PV fluctuations. Achleitner et al. deployed an online sensor network in rooftops of neighboring buildings for forecasting purposes. The system was based on IEEE 802.15.4/ZigBee protocol, comprised 19 motes, and operated at a sampling period of 5 s [23]. The network described in [26] was also based in the 802.15.4/ZigBee protocol. In this case, it comprised 16 nodes regularly distributed over 15 × 15 m. The authors found network instability for sampling periods shorter than 500 ms and with a certain lack of synchronization. In [28], five nodes implementing LoRa were used to test a nowcasting method for “grid-friendly control” of PV systems. The network is conceived as a partial prototype of a theoretical monitoring system composed of two concentric circles of sensors with radii of 500 and 440 m, respectively, comprising a total of 48 pyranometers. The partial prototype of five sensors is designed to cover cloud motion directions from 0${}^{\circ }$ to 60${}^{\circ }$ North. The system described in [24] is designed to monitor a utility scale PV system using National Instruments (NI) devices. It cannot strictly be considered an irradiance monitoring network, as it is focused on power measurements and only three of the 12 motes include pyranometers but illustrates the feasibility of deploying Wireless Sensor Networks (WSN) with the required specifications in PV plants to provide additional information for PV plant control. Some of these studies will be used for comparison in Section 5.

The present article deals with the design and test of a wireless irradiance monitoring network with specifications comparable to the wired ones regarding sampling periods, stability, and synchronization. The article is structured as follows: Section 2, Section 3, Section 4 and Section 5 and finalize with Section 6.

## 2. Materials and Methods

### 2.1. Wireless Protocols

The communication protocol is a fundamental element that strongly determines the design and specifications of a WSN. Since this WSN aims to support nowcasting applications for a wide range of PV systems, the design process will prioritize the achievement of high bandwidth and maximum spatial coverage as possible with reasonable power consumption and economic cost.

Bandwidth determines the amount of data that can be transmitted on the network per unit of time, thereby being a constraint for the maximum amount of motes for a specific sampling rate. In this sense, the selected wireless protocol aims to provide the maximum versatility in the amount of motes in the WSN and achievable sampling rate. On the other hand, coverage maximization pretends to provide large area connectivity using the minimum amount of wired parts, e.g., network devices that will require wired connection to a Local Area Network (LAN). Thus, the selection of the wireless protocol involves a trade-off between the two considerations, since high bandwidth and poor coverage are as detrimental as high range and inadequate bandwidth for the proposed objective. Table 2 is given to facilitate the comparison of main specifications of most commonly used wireless protocols.

Among these alternatives, Bluetooth was discarded due to its short range. As commented in the previous section, [26] found a maximum acceptable limit of 500 ms with 16 motes using ZigBee [26]. Since LoRa has lower data rate, it was also discarded. Finally, WiFi was the selected protocol for their range and data rate.

### 2.2. Application Layer Protocols

The Application Layer Protocol (ALP) is an abstraction layer that defines how applications should communicate. There are several ALP available for IoT messaging, but most commonly used ones are provided in Table 3.

According to the information on the table, MQTT is a good candidate for ALP attending to two considerations: low overhead (header size) and high usage as M2M/IoT protocol. Low overhead is important since the required amount of information for data transmission is minimized, while the high IoT usage is a guarantee of good support of the protocol. Furthermore, 3 levels of QoS (Quality of Service) is an additional valuable feature for ensuring data reception. In this study, a QoS of 0 will be used, which is the minimum.

### 2.3. Development Board

The board selected for the development of the WSN is based on the ESP32 microcontroller from Espressif Systems, which includes features such as a dual-core 32-bits microprocessor running up to 240 MHz, 12-bit ADC channels, several digital interfaces such I2C, SPI, Serial and built-in Wi-Fi and Bluetooth connectivity, all of which are highly valuable for online WSN. As negative aspects, ESP32 microcontroller has some known issues, the most notable being the high noise and nonlinearity of the built-in ADC.

In order to overcome this issue, the manufacturer recommends connecting a 0.1 μF capacitor to the ADC pad and multi-sampling to reduce the noise profile [37]. The nonlinearity issue was also addressed in [38], and authors concluded that a calibration is compulsory for reliable results or using a narrow range where the ADC behavior can be considered to be linear. We conducted both procedures, but the observed improvement was not as relevant as expected. Furthermore, it was noticed that the noise profile changed with the sampling rate.

This issue can be solved using an external ADC if noticeable noise reduction is achieved. For this purpose, the ADS1115 IC was chosen as ADC because it provides up to 4 channels of 16-bit precision and up to 860 Samples Per Second (SPS) over I2C, with promising low noise profile and can be found in ready to use PCB boards.

Figure 1 shows the performance comparison for 200, 100, 50, and 10 ms of a sampling period of both ADCs. Each sampling period was tested for a duration of 60 s. A triangular signal of 2 s of periodicity was used for testing purposes, covering the expected pyranometer output from 700 W/m${}^{2}$ to 800 W/m${}^{2}$, as a representation of a linear irradiance ramp of 100 W/m${}^{2}$ per second. The behavior of the function generator was considered ideal. The embedded ESP32 ADC was calibrated with a linear regression for this range in order to compute noise, excluding nonlinearity response which is another drawback for the use of the built-in ADC. The ADS1115 was configured with a data rate of 128 SPS, which is capable of achieving 10 ms of sampling rate without overlapping samples.

In Figure 1, each column represents the same sampling period, indicated at the top text. The first two rows correspond to the four seconds of the test, the upper one corresponds to the built-in ADC as labelled on the right side, and the second row corresponds to the ADS1115. The gray lines represent the ideal input waveform and the red dots the sampled values from ADC. Error histograms with 1 W/m${}^{2}$ of bin size are shown following the same organization in the last 2 rows.

The noise reduction provided by the ADS1115 over built-in ADC from histograms in Figure 1 is noticeable. The noise profile of the embedded ADC can be assumable for a single irradiance monitoring sensor, but it is a major issue when the 16 signals captured from motes are processed by cloud motion estimation algorithms. Based on the previously mentioned and the noise reduction achieved with the inexpensive external ADC, authors consider that its usage is justified.

### 2.4. Sensor Selection

Pyranometers can be classified into two main types according to their working principle: thermopiles and photodiodes. While thermopile based pyranometers have better spectral response than photodiode ones, they are more expensive and present longer response times. Photodiode based pyranometers have a response time on the order of milliseconds, while thermopile ones usually stand on the order of seconds. These differences make the photodiodes based pyranometers adequate for a low cost and high sample rate irradiance monitoring network. Based on the above statement, a good candidate sensor for monitoring solar irradiance is the SP-214, which is a photodiode pyranometer from Apogee Instruments, Inc. Logan, USA. Its main specifications are given in Table 4.

### 2.5. Mote Prototype

The mote was built by connecting the pyranometer to the external ADC (ADS1115), and the latter to the ESP32. The irradiance sensor and the voltage regulator are powered using an external 15V AC-DC power supply with additional voltage regulations inside the prototype box. The voltage regulator is a DC–DC step down, providing 5V at the output, which is acceptable to power the ADS1115 and ESP32 boards. Figure 2 illustrates the connections among the different components of the mote.

The electronic circuits of the prototype were placed inside a box (Figure 3a), except for the irradiance sensor, which was placed outside connected by a 5 m cable (Figure 3b).

For the load resistance, $R_{L}$, a 150 Ω value resistor, was used to convert the 4–20 mA output from pyranometer to voltage ranging from 0.6 to 3 V. Based on the electronic diagram of Figure 2, the ADS1115 input impedance creates a current divider, and its effects should be quantified.

Texas Instruments characterize the typical ADS1115 IC input impedance in 6 MΩ for the common-mode, with a sensibility of 125 μV/count and 8 SPS. Texas Instruments does not provide electrical characterization for data rates different to 8 SPS. This study tested the ADS1115 with 8 SPS and 128 SPS, and no difference was found. Hence, it can be assumed for current divider estimation that the input impedance stays on the order of 6 MΩ. On this basis, the deviation of voltage drop in $R_{L}$ should be much less than 0.01%, which is small enough to be assumed with the benefit of no extra circuitries like operational amplifiers in voltage follower configuration.

### 2.6. Server

The WSN requires a server and a software stack to perform the data processing and store the irradiance values sent by the motes. The software was intended to be lightweight and the bare minimum. Initially, the software stack consists of an MQTT Broker, databases with agents for processing and pushing data to databases. Mosquitto is the software chosen as an MQTT Broker due to its extended usage and its open source license. For data storage, two databases were tested: InfluxDB, which is a time series database, and MariaDB, which is a relational one, are both open source.

Some agent is needed between the MQTT Broker and the databases. For that purpose, the Telegraf agent was chosen for collecting data from the MQTT broker, processing and pushing them into InfluxDB (a simplified configuration file is provided in Code 1, Appendix A), while, for storing data in the MariaDB, the Node-RED and the data flow shown in Figure 4 were used.

Figure 5 offers a visual description of the network topology initially used.

The hardware used in this study was an HP Proliant MicroServer Gen8 (HP Inc., Palo Alto, CA, USA) with a XEON E3160L and 12 GiB of RAM at 1333 MHz. The required software was installed on virtualized Debian 11 running on a LXC container in Proxmox VE 7.1. from Proxmox Server Solutions Gmbh (Wien, Austria). Some performance loss could be expected due to virtualization technology.

### 2.7. Performance Test Design

The performance test is focused on characterizing the network stability, data synchronization, and storage process efficiency. To achieve this goal, the network motes were programmed using timer interruption at a fixed rate to prevent time drifting due to processing time. The interruption period can be changed remotely using a specific MQTT topic that reprograms the timer according to the requested period. In each timer interruption, the microcontroller sends Unix time with millisecond precision, providing a timestamp to contextualize the samples. The capability of each mote to perform sampling at a fixed rate can be assessed by computing the vector *D*, defined as the differences of sent times (*T*) of consecutive measurements (k,k−1) from one mote: (1)D[k−1]=T[k]−T[k−1]∀k∈[1,2,3...n]

On the server side, each datum is stored with the reception timestamp with the precision of milliseconds. Similarly to Equation (Equation 1), the reception stability can be analyzed using time differences of a reception timestamp.

Finally, packet loss ratio, defined as the amount of packets not received over the expected, is another relevant aspect to characterize the network stability. Calling $T_{test}$ the test time for the sampling period $T_{s}$, the number of expected packets per device, $p_{e}$, can be calculated as: (2)$$p_{e}=\frac{T_{test}}{T_{s}}$$

Let $p_{r}$ be the amount of packet received from one device during $T_{test}$ time and sampling period $T_{s}$, the packet loss ratio, $p_{l}$, can be defined as follows: (3)$$p_{l}\left(\%\right)=\frac{p_{e}-p_{r}}{p_{e}}\cdot 100$$

As stated above, operational data for PV plants and nowcasting techniques require sub-second sampling. Based on this, the performance test was designed to analyse the following sampling periods: 500, 300, 200, 100, 50, 25, and 10 ms, with a duration of 600 s per sampling period.

Since ESP32 has the capability to perform time synchronization using Simple Network Time Protocol (SNTP) against a NTP server [39], this functionality will be used for a timestamp each datum with absolute time reference for better traceability. In this case, the Masterclock GMR5000 (Figure 6) has been used as a precision local NTP server for accurate characterization purposes, but a public or another kind of local NTP server could be used. Each mote performs a NTP synchronization after the boot up, and sets up the timer for interruptions with the UTC second changes. Under normal operation mode, NTP resynchronization is done once every 10 min to avoid noticeable time drift of the RTC timer.

In the preliminary implementations of the performance test, it was evident that the Node-RED and MariaDB branch did not work as expected: data ingestion demanded an excessively long time at sampling periods shorter than 200 ms. Consequently, for the sake of simplicity in testing and presentation of results, this branch was discarded and hence the topology shown in Figure 7 was used in the following sections.

## 3. Network Performance Test Results

As stated above, sampling stability, reception stability, and packet loss ratio will be analyzed. The results are presented grouped by sampling period, providing a characterization of the performance of the WSN for each working state. Figure 8 corresponds to the sampling histograms of 16 motes.

It can be inferred from the histograms of Figure 8 that the WSN motes present an excellent sampling behavior. For each sampling period, a mean deviation of 0 ms is observed thanks to the timer interrupt that avoids time drifting.

A good mote sampling stability does not imply reception stability, most especially in wireless devices. Following the procedure described in the previous section, the reception stability was analyzed based on the server reception time differences between two consecutive packets for the same device. Figure 9 shows the histograms associated with this analysis by aggregating the individual results of the 16 motes for each sampling period.

The histogram associated with the sampling period of 10 ms in Figure 9 shows a bimodal distribution. From that distribution, it can be inferred that delayed packets overlap in time with the next ones; hence, it cannot be assumed that the reception time is representative of the sample time when sampling periods are in this range.

For all the other sampling period histograms, the reception time deviation stays in a range of ±10 ms, with the highest peak at the sampling period interval bin and almost symmetrical distribution considering a bin size of 1 ms. As a special particularity, the highest probability to receive the data at requested interval ±0.5 ms is given for a sampling period of 50 ms, followed by 25 ms. These results can be counter-intuitive since it reflects a better performance at these sampling periods than at longer sampling periods, when the network is supposed to be less congested.

Another important performance aspect to analyze is the packet loss ratio. This ratio provides a quantization of packets that did not reach the end of the data flow, which is in this case the database. The cause of the packet losses could be diverse: packet not sent by the mote, mote disconnections, packet lost by the WiFi network, or even data not processed properly by the server. Figure 10 shows packet loss ratio versus sampling period.

Attending to the Figure 10, no correlation was found between sampling period and packet losses. Packet loss ratio was kept under 0.45% for all tested sampling periods, thereby being a low value with no relevant impact for WSN purposes.

## 4. Irradiance Field Results

The developed WSN was deployed at the rooftop of the University building in a regular 4-by-4 gridded configuration with a minimum inter-sensor distance of 5 m, covering an area of 15 × 15 m, as can be seen in Figure 11. The red circles of the figure indicate the pyranometers’ positions, while the blue one locates the router and power supply.

A monitoring campaign was done using the presented distribution on 10 February 2022 under partial cloud cover conditions. For this campaign, a sampling period of 50 ms was used—half of the 100 ms period sufficient to capture the fastest irradiance changes [22,33]—to offer a comparative of longer sampling periods. The acquired GHI time series’ values include several interesting cloud passages for analysis. One of them is represented in the 20 s time-series shown in Figure 12, which started at 12:40:22 UTC time.

The same irradiance values can be represented in a 4-by-4 matrix for each sampling time, representing the spatial distribution. This technique provides a visual representation of the irradiance field evolution, allowing a better comparison between sampling periods. Using the previous 20 s irradiance time-series, the 2D representation is shown in Figure 13. Snapshots for sampling periods of 1000, 500, 200, and 100 ms are also represented using a subsampling technique to offer a visual comparison of the evolution for several sampling periods.

Note that, for shorter sampling periods, the irradiance field evolution is captured with high detail but requires more snapshots to capture a complete ramp.

The continuation of the above time series with a total duration of one minute is shown in Figure 14. Snapshots evolution using these data and downscaling to a sampling period of 500 ms can be seen in Figure 15.

The stated results reveal the potential of the developed WSN to online monitor the irradiance field at fast sampling periods. This makes this WSN a valuable data source for PV power plants with the benefits in costs of a wireless alternative and for research to apply cloud shadow motion algorithms and nowcasting techniques.

## 5. Discussion

This section aims to contextualize the results of the previous two sections in relation to other studies. Assuming a certain quality of pyranometers, the key factor in evaluating the usefulness of an irradiance sensor network for nowcasting applications is the sampling period. In this aspect, the results described in Section 3 illustrating adequate performance for 50 ms—even 25 ms—can only be compared to the 40 ms in [22], and significantly outperforms the sampling rates of the WSN summarized in Table 1. In any case, the WSN is able to sample at 100 ms, which which is considered sufficient to describe the fastest changes of irradiance [22,40].

Another fundamental aspect is the capability of the network of supporting cloud motion estimation algorithms, which is critical information for irradiance nowcasting systems. Espinosa-Gavira et al. defined the relationship between sampling period, sensing area, and cloud shadow speed in [41]. This and other studies established 30 m/s as an upper limit of cloud shadow speeds for testing the operation of PV systems, since higher speeds have an extremely low occurrence [40]. Applying the equations, a cloud passage at the maximum expected speed, 30 m/s, will be registered in 5 snapshots using a 15 × 15 m gridded configuration and a sampling period of 100 ms, which should be enough to support a nowcasting system. In this sense, larger area networks and/or lower cloud speeds will be registered in more snapshots. In summary, a sampling period of 100 ms should also be sufficient for most of the monitoring systems if there are cloud motion estimation algorithms.

The WSN showed great stability of sampling and reception with a very low packet loss ratio. The motes were kept in synchronization via the NTP server, avoiding the time drift of the RTC, and enabling the contextualization of samples with absolute timestamping. It would be interesting to compare these results with other studies, but this type of characterization is generally neglected in similar research. In this sense, it would be desirable that further studies incorporate characterizations of sampling, reception, synchronicity, or packet losses to determine which is the most efficient technology for this problem.

The use of WSN to provide additional information for PV system control has proven to be a feasible solution using NI-WSN [24] or ZigBee [42], although, in these cases, the networks were mainly focused on power measurements. According to the results in [26], the application of these technologies—both based in the IEEE 802.15.4 standard—to irradiance measurement would encounter problems in achieving sampling periods of 100 ms and the required level of synchronization. In fact, in [24], the IEEE 1588 Precision Time Protocol (PTP) standard is used to guarantee synchronization, with a consequent increase in system complexity, but maintaining a 10 s sampling period. LoRaWAN could represent another communication option, but its maximum data rate is below the IEEE 802.15.4 standard, being only able to achieve sub-second sampling under very favorable conditions [43]. In this framework, the use of WiFi with NTP synchronization represents an adequate option to achieve the requirements of an irradiance monitoring system for nowcasting applications, with options of scalability by adding new routers without performance penalties.

The ESP32 has demonstrated to be an IoT device capable of performing outdoor monitoring using WiFi [44,45]. For two weeks, the WSN was deployed on the university building rooftop under variable weather conditions, showing no appreciable difference in performance.

## 6. Conclusions

This study deals with the design and testing of an online wireless irradiance sensor network capable of providing real-time data to support PV systems operations and irradiance nowcasting. The WiFi protocol has been chosen with the objective of maximizing spatial coverage with sufficient bandwidth to achieve the most demanding requirements of research irradiance monitoring networks, while providing flexibility to add new motes or spatially move existing ones without economic impact. The designed wireless network proved its capability to achieve sampling periods up to 25 ms with good coordination and very low packet loss ratio, thus clearly outperforming the period of 100 ms that is considered enough to detect the fastest expected irradiance phenomena. Irradiance data of a monitoring campaign was spatially represented, showing that the system was able to capture detailed features of the irradiance field and its evolution. It was also discussed how the network measurement capabilities should be enough to capture even the fastest clouds and to be adapted to different extensions. The availability of this information with a low latency makes this monitoring system a valuable tool for a wide range of PV systems control and future research. Analysis of the mote side showed that the devices have enough computing power to overpass the most demanding test, as shown in Figure 8. On the server side, samples have a good behavior for a sampling period of 25 ms with good stability and packet loss ratio lower than 0.45% Based on the test conducted and results, the designed WSN is capable of achieving comparable specification to a state-of-the art irradiance monitoring network with the advantages in costs of the wireless technology.

Future work will improve the designed WSN by replacing the external power supply for a small PV panel and a battery pack to provide fully autonomous wireless features. Data collected by the WSN will be used to apply cloud motion estimation algorithms, nowcasting techniques, and how collaborative nearby irradiance networks could improve forecasting skills.

## Figures and Tables

**Figure 1 sensors-22-02928-f001:**
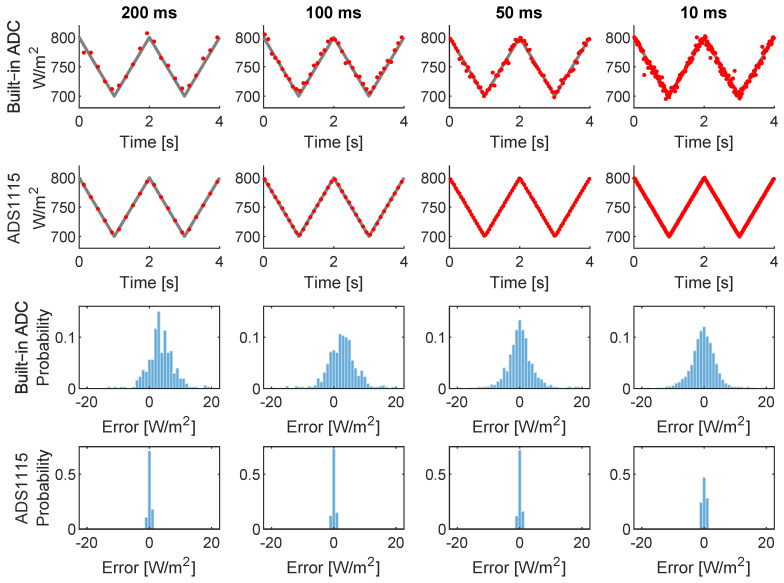
Comparison between ESP32 embed ADC and the external ADC ADS1115.

**Figure 2 sensors-22-02928-f002:**
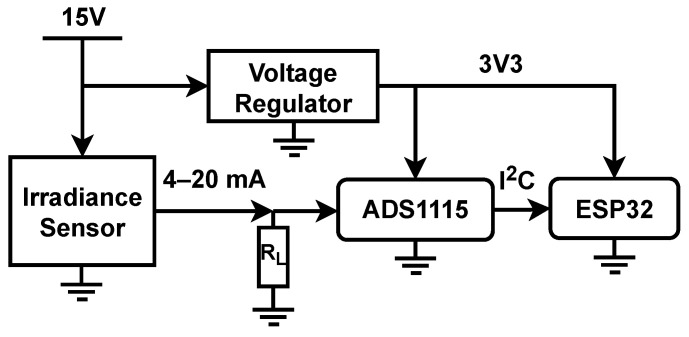
Electronics box diagram of one mote of the WSN.

**Figure 3 sensors-22-02928-f003:**
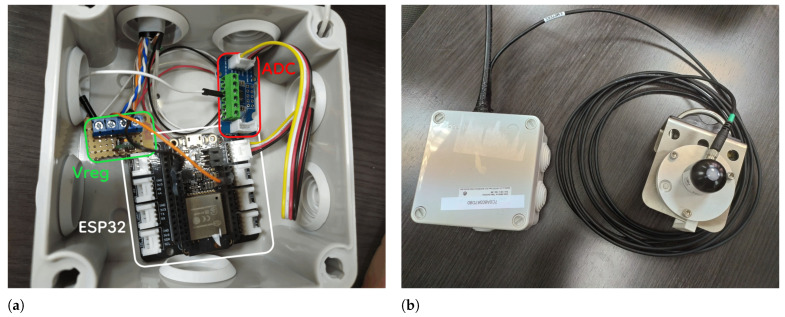
Inside and outside looking of the mote prototype. (**a**) Elements inside the box with Voltage Regulator (marked as Vreg); (**b**) box and pyranometer.

**Figure 4 sensors-22-02928-f004:**
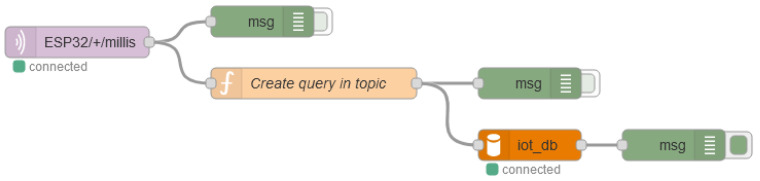
Node-RED data flow.

**Figure 5 sensors-22-02928-f005:**
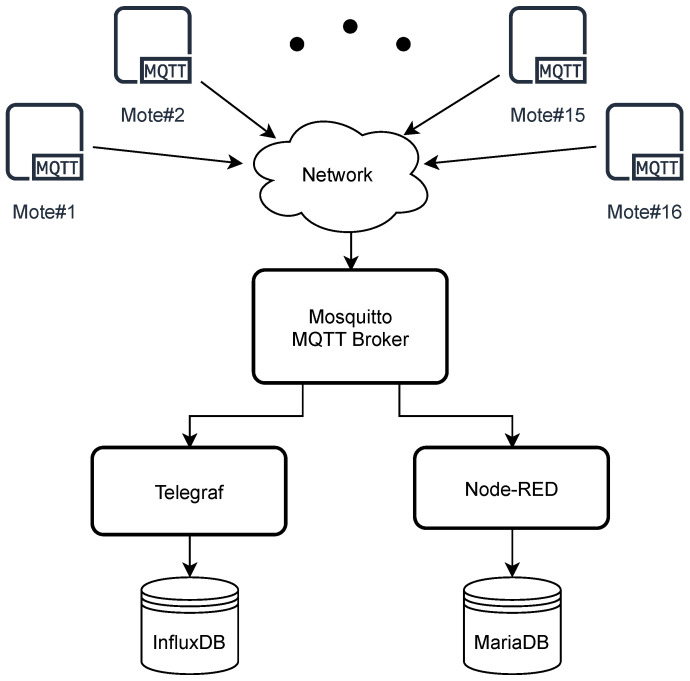
Initial network topology. Black dots refer to other motes devices.

**Figure 6 sensors-22-02928-f006:**
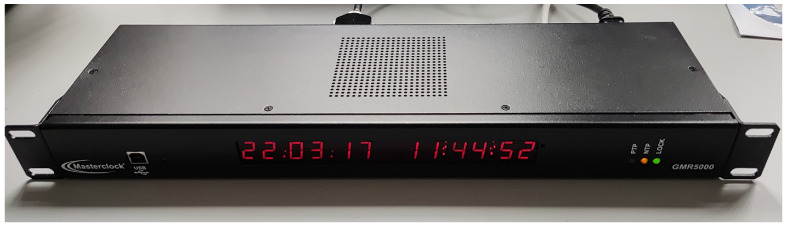
GMR5000 (from Masterclock Inc., St Charles, MO, USA) as an NTP local server used for time syncing.

**Figure 7 sensors-22-02928-f007:**
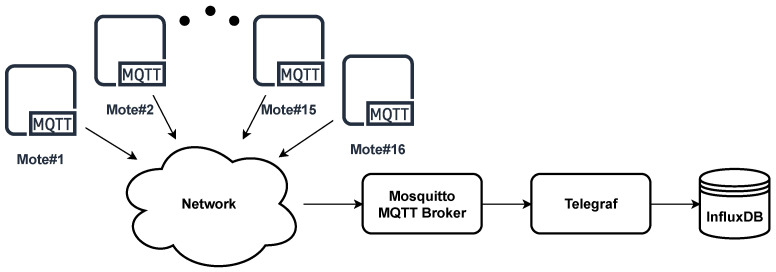
Final network topology.

**Figure 8 sensors-22-02928-f008:**
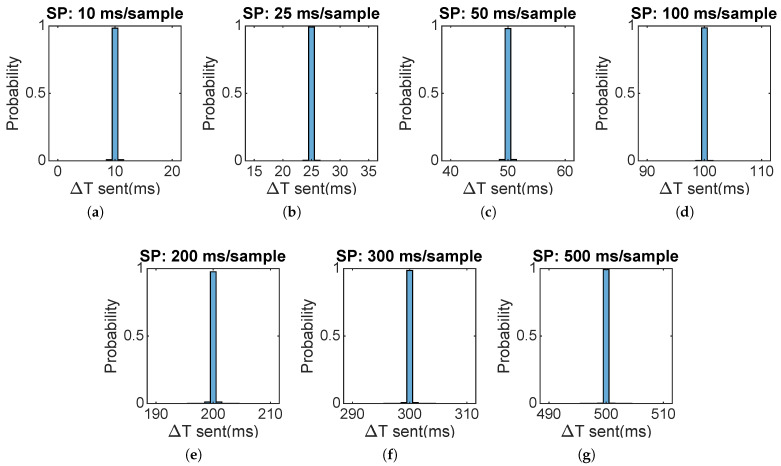
Sampling histograms for each sampling period tested. Each Sampling Period (SP) is indicated in the histogram title (**a**–**g**).

**Figure 9 sensors-22-02928-f009:**
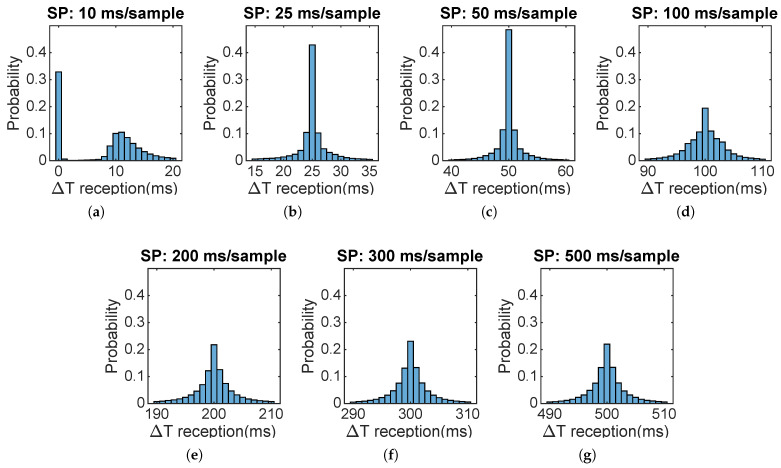
Histogram of reception time difference for each sampling period, indicated in the histogram title (**a**–**g**).

**Figure 10 sensors-22-02928-f010:**
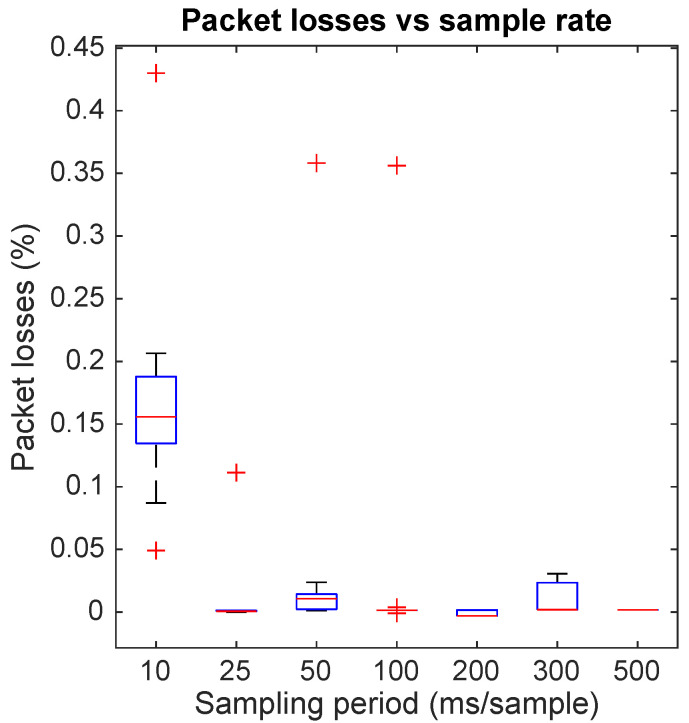
Packet loss ratio versus tested sampling periods.

**Figure 11 sensors-22-02928-f011:**
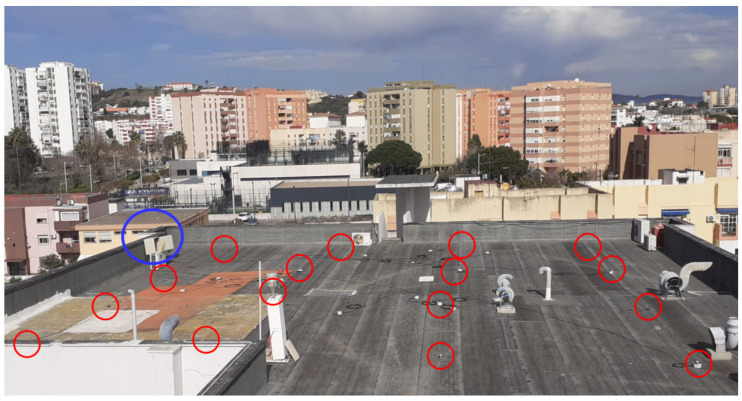
WSN deployed at the rooftop of the University building. Red circles indicate pyranometers, and the blue circle locates the position of the router and AC–DC power supply.

**Figure 12 sensors-22-02928-f012:**
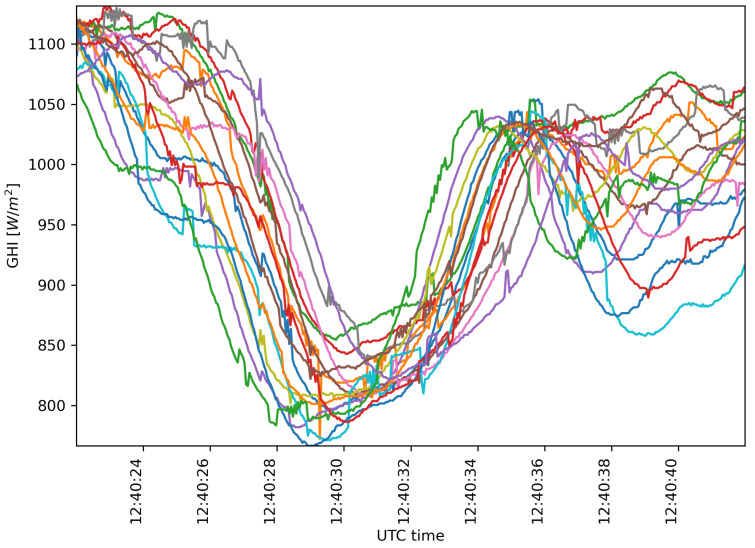
20 s of 16 irradiance time series captured from WSN on 10 February 2022 starting at 12:40:22 UTC time.

**Figure 13 sensors-22-02928-f013:**
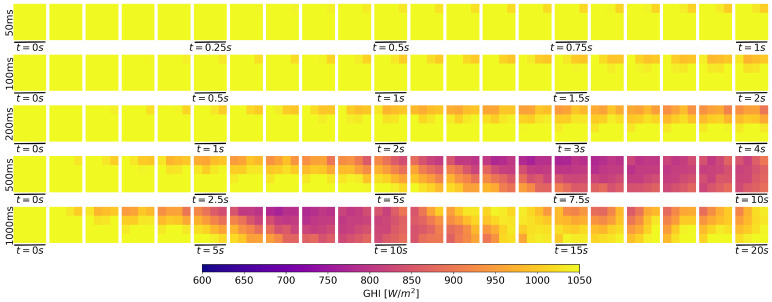
Snapshots evolution from data represented in Figure 12. Each row represents the evolution at different sampling periods.

**Figure 14 sensors-22-02928-f014:**
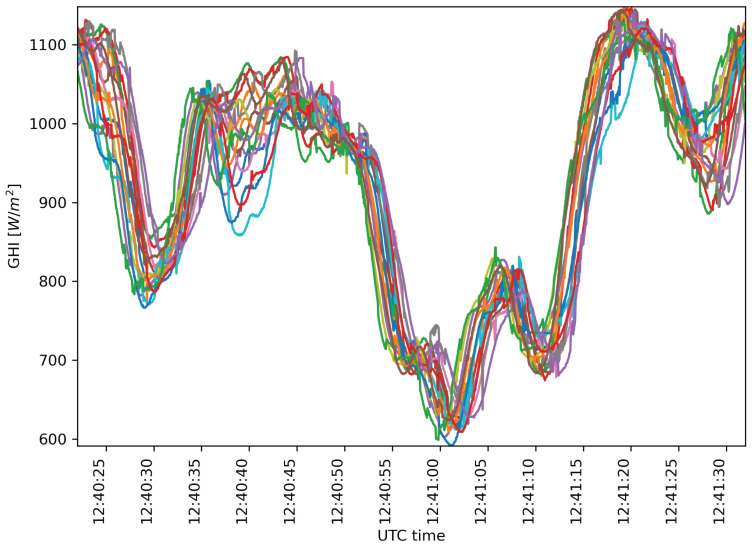
60 s of 16 irradiance time series captured from WSN on 10 February 2022 starting at 12:40:22 UTC time.

**Figure 15 sensors-22-02928-f015:**
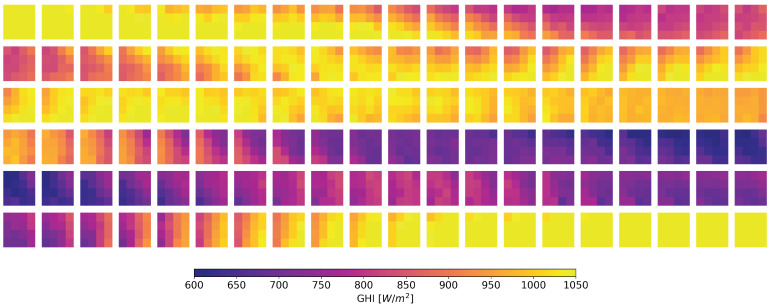
60 s of snapshots evolution at 500 ms from time series shown in Figure 14 acquired with the designed WSN. Each row represents 10 seconds of the evolution.

**Table 1 sensors-22-02928-t001:** Irradiance sensor networks.

Ref.	Year	Sampling Period	N Pyran.	Area (Approx.)	Online?/Technology?
[22]	2013	40 ms	21	50 m × 50 m	No
[23]	2014	5 s	19	400 m × 500 m (5 buildings)	Yes/IEEE 802.15.4—ZigBee
[21]	2015	1 s	20	350 m × 280 m (gridded)	No
[24]	2016	10 s	3	200 m × 130 m	Yes/IEEE 802.15.4—NI WSN
[20]	2017	1 s	48	1.5 km × 1.5 km (gridded)	No
[25]	2017	1 s (avg 1 min)	24	42 km × 23 km	No
[26]	2018	0.5 s	16	15 m × 15 m (gridded)	Yes/IEEE 802.15.4—ZigBee
[27]	2019	1 min	2	15 km (distance)	Yes/WiFi+IoT
[28]	2019	1 s	5	200 m × 200 m (5 buildings)	Yes/LoRa
[29]	2020	5 min	10	100 m × 50 m	Yes/IEEE 802.15.4 MAC
[19]	2021	1 s	50	2 km × 2 km	No
[30]	2021	0.1 s	21	25 m × 50 m	No
[31]	2022	1 s (avg 1 min)	40	200 m × 200 km	Yes/Mobile network
[18]	2022	1 s	17	1 km × 1 km	No

**Table 2 sensors-22-02928-t002:** Comparative of main Wireless IoT communication protocol: Bluetooth, LoRa, ZigBee, and Wi-Fi. Simplification made from [36]: Table 2.

Technology	Bluetooth	LoRa	ZigBee	WiFi
Standard	IEEE 802.15.1	LoRaWAN	IEEE 802.15.4	IEEE 802.11 a/c/b/d/g/n
Frequency band	2.4–60 GHz	868/900 MHz	868/915 MHz; 2.4 GHz	2.4–60 GHz
Data rate	1–24 Mb/s	0.3–50 kb/s	40–250 kb/s	1 Mb/s–6.75 Gb/s
Typical range	10 m	30 km	10–100 m	100 m

**Table 3 sensors-22-02928-t003:** Comparison of main ALP for IoT messaging.

Specification∖Protocol	MQTT	CoAP	AMQP	HTTP
Transport layer	TCP	UDP	TCP	TCP
Min. header size	2 Bytes	4 Bytes	8 Bytes	26 Bytes
QoS/Reliability	3 Levels	2 Levels	2 Levels	Limited
M2M/IoT usage	Largest	Medium	High	Low

**Table 4 sensors-22-02928-t004:** Main specifications of the pyranometer SP-214 (Apogee Instruments, Inc., Logan, UT, USA).

Specification	Value
Power supply range	7–24 V DC
Output range	4–20 mA
Calibration factor	125 W/m${}^{2}$ per mA + 4 mA offset
Sensibility	0.008 mA per W/m${}^{2}$
Measurement range	0–2000 W/m${}^{2}$
Nonlinearity	<1% (up to 2000 W/m${}^{2}$)
Response time	<1 ms

## Data Availability

Not applicable.

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
