# Peer review of "Design and Test of a High-Performance Wireless Sensor Network for Irradiance Monitoring"

_sensors, 2022, doi:10.3390/s22082928_

Round 1

Reviewer 1 Report

This manuscript presents "Design and Test of a High-Performance Wireless Sensor Network for Irradiance Monitoring". However, the following observations are made:

  1. Previous studies are missing. Add all related studies and compare all in a table.
  2. Figure 6 is repeated two times. Correct all figure numbers.
  3. Discuss the results and compare them to previous studies in a separate discussion section.
  4. Add latest references from 2021/22.

Author Response

Please see the PDF attachment

Reviewer 2 Report

Please, see the attached PDF file.

Reviewer 3 Report

To: Sensors

Manuscript ID: Sensors - 1663936

This research focuses on designing and testing a wireless irradiance monitoring network with specifications comparable to the wired ones regarding sampling periods and time synchronization. The work is meaningful for monitoring systems, and the paper is well written and presented. There are several issues the authors need to address before the manuscript gets published in sensors. They are listed as follow:

  • The introduction has covered the most relevant works in the field. I would recommend the authors add a table that summarizes the advantages and disadvantages of the existing sensors and what is the added value of this research compared to them.
  • The sentence in line 94 can’t stand alone.
  • The WSN is designed to cover a wide bandwidth, what is the proposed range?
  • Please organize section “2.6. Server”.
  • It is recommended to add an appendix and move the code presented in section 2.6 to it.
  • “The performance test is focused on characterizing the network stability, data synchronization, and storage process efficiency.” Have the authors considered the external disturbance due to the wind and any external source of vibration?
  • How did you ensure that the monitoring sensors are running and synchronized at the same time? There should be some drifting?
  • Merge paragraphs in lines 233-236 and make it as one paragraph.
  • Add (,) after From that distribution.
  • Are the sensors oriented on top of the building? If yes, what would happen if they were placed randomly?
  • How about the lifetime of the monitoring systems especially if it is placed in a hard environment?

Round 2

Reviewer 1 Report

This manuscript is revised according to my previous comments.  Now, I am agree to accept it in its present form.

Reviewer 2 Report

No comments added